# Safety and Efficacy of the Transaxillary Access for Minimally Invasive Aortic Valve Surgery

**DOI:** 10.3390/medicina59010160

**Published:** 2023-01-13

**Authors:** Manuel Wilbring, Konstantin Alexiou, Torsten Schmidt, Asen Petrov, Ali Taghizadeh-Waghefi, Efstratios Charitos, Klaus Matschke, Sebastian Arzt, Utz Kappert

**Affiliations:** 1Center for Minimally Invasive Cardiac Surgery, University Heart Center Dresden, 01307 Dresden, Germany; 2Department of Cardiac Anesthesiology, University Heart Center Dresden, 01307 Dresden, Germany; 3Department of Cardiac Surgery, Kerckhoff Heart Center, 61231 Bad Nauheim, Germany

**Keywords:** aortic valve replacement, minimally invasive surgery, transaxillary

## Abstract

*Background and Objectives*: Transaxillary access is one of the latest innovations for minimally invasive aortic valve replacement (MICS-AVR). This study compares clinical performance in a large transaxillary MICS-AVR group to a propensity-matched sternotomy control group. *Materials and Methods*: This study enrolled 908 patients undergoing isolated AVR with a mean age of 69.4 ± 18.0 years, logistic EuroSCORE of 4.0 ± 3.9%, and body mass index (BMI) of 27.3 ± 6.1 kg/m^2^. The treatment group comprised 454 consecutive transaxillary MICS-AVR patients. The control group was 1:1 propensity-matched out of 3115 consecutive sternotomy aortic valve surgeries. Endocarditis, redo, and combined procedures were excluded. The multivariate matching model included age, left ventricular ejection fraction, logistic EuroSCORE, pulmonary hypertension, coronary artery disease, chronic lung disease, and BMI. *Results*: Propensity-matching was successful with subsequent comparable clinical baselines in both groups. MICS-AVR had longer skin-to-skin time (120.0 ± 31.5 min vs. 114.2 ± 28.7 min; *p* < 0.001) and more frequent bleeding requiring chest reopening (5.0% vs. 2.4%; *p* < 0.010), but significantly less packed red blood cell transfusions (0.57 ± 1.6 vs. 0.82 ± 1.6; *p* = 0.040). In addition, MICS-AVR patients had fewer access site wound abnormalities (1.5% vs. 3.7%; *p* = 0.038), shorter intensive care unit stays (*p* < 0.001), shorter ventilation times (*p* < 0.001), and shorter hospital stays (7.0 ± 5.1 days vs. 11.1 ± 6.5; *p* < 0.001). No significant differences were observed in stroke > Rankin 2 (0.9% vs. 1.1%; *p* = 0.791), renal replacement therapy (1.5% vs. 2.4%; *p* = 0.4762), and hospital mortality (0.9% vs. 1.5%; *p* = 0.546). *Conclusions*: Transaxillary MICS-AVR is at least as safe as AVR by sternotomy and can be performed in the same time frame. Its advantages are fewer transfusions and quicker postoperative recovery with a significantly shorter hospital stay. The cosmetic result and unrestricted physical abilities due to the untouched sternum and ribs are unique advantages of transaxillary access.

## 1. Introduction

Minimally invasive aortic valve surgery (Minimally Invasive Cardiac Surgery—Aortic Valve Replacement (MICS-AVR)) is not a trademark. It rather is an umbrella term encompassing a heterogeneous group of access routes, cannulation strategies, and surgical techniques.

Since Rao and Kumar first reported a minimally invasive aortic valve replacement in 1993 and Cosgrove and Sabik’s first small case series in 1996, minimally invasive aortic valve techniques have evolved continuously [1,2].

While Rao and Cosgrove used different surgical access routes, Cosgrove described something conceptually new, combining femoral cannulation with a reduced thoracal incision for more space and surgical convenience at the thoracic access site [2].

However, both of these pioneering access routes, right thoracotomy (Rao et al.) and parasternal thoracotomy (Cosgrove et al.), have been left behind, and newer access routes have been established for MICS-AVR [1,2].

The most widespread access is the mini- or hemi-sternotomy. Notably, the initially described “J” ministernotomy was a manubrial-sparing sternotomy [3]. However, the abstract concept changed over time, and ministernotomy currently means a manubrial-cutting technique. Many variants of the ministernotomy have been developed, denoted using other letters of the alphabet (“J,” “I,” “H,” inversed “T,” and reversed “C” sternotomy), sometimes sawing into the third, fourth, or even fifth intercostal space [3,4,5,6,7]. Nevertheless, this ministernotomy approach in all its forms remains a bone-breaking trauma for the patient.

Joseph Lamelas is a proponent of a bone-sparing alternative with his Miami method, better known as right anterior thoracotomy (RAT), for minimally invasive aortic valve surgery [8]. This technique uses a RAT to access the second intercostal space. Extracorporeal circulation (ECC) is established via the femoral vessels, the third rib is dissected from the sternum, and the right mammary artery is mostly sacrificed [8]. The advantage of preserving the rib is obvious. However, sacrificing the right mammary artery and possible rib instability or lung herniation are apparent disadvantages. To further reduce this trauma, an endoscopic variation was described by Van Praet et al. in 2020 [9].

In 2021, our group established a simplified “single incision—direct vision” concept with transaxillary access [10]. Transaxillary access for isolated minimally invasive aortic valve surgery includes a 5 cm skin incision in the right anterior axillary line, accessing the third or fourth intercostal space. ECC is established using standard femoral cannulation, and the operative field is brought into an operable distance using pericardial stay sutures [10]. This technique can be described as technically less demanding, characterized by an easy setup, short procedure times, and excellent initial outcomes with low rates of adverse events [10]. However, the knowledge of this novel technique’s safety and efficacy is based only on its initial description, including a small series of the first 50 patients [10].

In the steadfast belief that sacrificing patient safety in favor of limited skin incisions is unconscionable and that minimally invasive surgery must at least be as safe as conventional surgery, this study compares the safety and efficacy of transaxillary access to conventional sternotomy in a large propensity-matched patient cohort.

## 2. Materials and Methods

### 2.1. Inclusion and Exclusion Criteria

This study included adult patients undergoing isolated aortic valve surgery, either minimally invasive using transaxillary access (treatment group) or conventional sternotomy (control group). Exclusion criteria were redo surgeries, combined procedures, including concomitant ablation or closure of the left atrial appendage, and active or recent endocarditis. The inclusion and exclusion criteria were chosen to narrowly define as homogenous a patient cohort as possible.

### 2.2. Study Design and Ethical Statement

This study is a retrospective observational cohort analysis of consecutive patients undergoing minimally invasive aortic valve surgery according to the inclusion criteria. A 1:1 propensity score matched control group was obtained from a retrospectively analyzed cohort of consecutive patients undergoing isolated aortic valve surgery by full sternotomy according to the inclusion criteria. Data were collected from the hospital database. This study was reviewed and approved by the local Ethic Board (EK—Nr. 298092012).

### 2.3. Patients and Groups

The groups were created from 5978 patients treated between January 2000 and May 2022 who fulfilled the inclusion criteria of isolated aortic valve surgery and met none of the exclusion criteria of endocarditis and redo surgery. Since transaxillary access was introduced in 2019, 454 consecutive patients undergoing surgery using this access were enrolled up to May 2022. This group served as the treatment group. Before 2015 and the change in access route policy, the proportion of minimally invasive access routes accounted for <2.0% at our institution. From 2015 to 2022, this rate increased to 96.3%. Only sternotomy patients from the pre-MICS era (2000–2014) were considered to reduce any potential undetected surgical selection bias.

Out of these 3115 consecutive patients, a 1:1 propensity-matched cohort of 454 patients served as the control group. The final study group comprised 908 patients, including 454 transaxillary and 454 sternotomy patients with mean age of 69.4 ± 18.0 years, logistic European System for Cardiac Operative Risk Evaluation (EuroSCORE) of 4.0 ± 3.9%, and body mass index (BMI) of 27.3 ± 6.1 kg/m^2^.

The transition period from 2015 to 2018 was excluded due to an obvious selection bias. During this period, the predominant access route for MICS-AVR was the RAT, with a series of 653 consecutive patients. This access was replaced by the transaxillary access beginning from 2019.

### 2.4. Statistical Analysis

The propensity score matching used a multivariate logistic regression model including age, sex, BMI, logistic EuroSCORE, coronary artery disease, pulmonary hypertension, chronic lung disease, and stratified left ventricular ejection fraction (LVEF). The treatment and control groups were 1:1-propensity matched based on these parameters. Figure 1 shows the covariate balance before and after adjustment.

Continuous data are presented as means with standard deviation. Categorial data are presented as absolute numbers and percentages. Intergroup comparisons of demographic variables used a *t*-test, Mann–Whitney-U test, Chi-square test, or Fisher’s exact test, as appropriate. Time-to-event outcomes were examined using the Kaplan–Meier method. A two-sided *p* < 0.05 was considered statistically significant. The statistical analyses were performed using SAS JMP 12.2 (SAS Institute, Cary, NC, USA) and the R statistical software (version 4.2.1; R Foundation for Statistical Computing).

### 2.5. Surgical Techniques

General anesthesia was used in all patients regardless of the surgical approach. Intraoperative transesophageal echocardiography was also performed as standard imaging monitoring in all patients.

#### 2.5.1. Treatment Group—Transaxillary Minimally Invasive Aortic Valve Surgery

Anesthesiologic specifics in this group included possible single lung ventilation and preoperative transvenous temporary pacing wire placement. The patient was placed supine with a slightly elevated right side of the chest. Their right arm was lifted into a javelin-thrower position and fixed to a surgical table arm support. The surgical access was performed as previously described [10,11].

Briefly, a 5 cm skin incision was made in the right anterior axillary line to open the third or fourth intercostal space. ECC was established through right femoral access after standard surgical cut-down. After establishing ECC, the pericardium was opened, and pericardial stay sutures were placed. After x-clamping of the aorta using a flexible Cosgrove clamp, crystalloid cardioplegia was administered for cardiac arrest. The aorta was opened by a transverse or longitudinal aortotomy depending on the operating surgeon’s preference and implanted valve type, enabling a perpendicular view of the aortic valve in direct vision. Specific equipment needed was a headlight, soft tissue retractor, and minimally invasive instruments. Figure 2 shows the intraoperative setup and the postoperative cosmetic result.

#### 2.5.2. Control Group—Full Sternotomy Aortic Valve Surgery

Full sternotomy in cardiac surgery is a well-known procedure. Briefly, complete median sternotomy and subsequent pericardial opening were performed in the usual manner. Cardiopulmonary bypass was established by cannulating the ascending aorta and the right atrium using a two-stage cannula. Antegrade crystalloid cardioplegia was administered via the ascending aorta. The left ventricular venting line was placed via the right superior pulmonary vein. The aortotomies used were (I) longitudinal, hockey stick-shaped or (II) transverse.

#### 2.5.3. Prosthesis Choice

The prosthesis choice generally depended on the operating surgeon’s preference. Especially in the MICS era and in the MICS-AVR patients, detailed anatomical information for procedural planning was gathered preoperatively by a high-resolution full cardiac cycle computerized tomography scan of the heart and vessels (TAVI-CT protocol). This information helped determine the expected distances from the annular plane to the chest wall, aortic annulus size, and the expected implanted valve size. This information was used in the surgeon’s decision-making process. Since increasing evidence supports the durability of the mainly implanted rapid deployment valve, their use was unrestricted [12,13,14].

#### 2.5.4. Intensive Care Unit Setup

The intensive care unit (ICU) was a 24-bed, highly standardized cardiac surgery-only ICU. One anesthesiologist has continuously overseen its medical management since 1997, employed as a consultant at the Department of Cardiac Surgery. Therefore, all processes, especially ventilation, patient blood, and hemodynamic management, were characterized by high standardization and continuity with minimal variations over time.

## 3. Results

### 3.1. Propensity Matching Results

Propensity score matching was performed based on the following parameters: age, BMI, LVEF by group, and logistic EuroSCORE. The matching process was successful, and no significant differences remained between groups in the matching parameters (Table 1).

### 3.2. Clinical Baseline Parameters

The final study group of 908 patients, with 454 patients in each group, was predominantly male (*n* = 567/908 (62.4%)) and had a mean age of 69.4 ± 18.0 years with an average body mass index (BMI) of 27.3 ± 6.1 kg/m^2^. Both the logistic EuroSCORE (4.0 ± 3.9%) and the EuroSCORE II (1.56 ± 1.0) indicated a low-risk population. Most patients had a preoperatively normal LVEF (>50%; *n* = 741/908 (81.6%)) and reported clinical dyspnea in New York Heart Association (NYHA) class III or IV (*n* = 577/908 (63.5%)).

Common comorbidities were diabetes mellitus type 2 (*n* = 257/908 (28.3%)), coronary artery disease (*n* = 248/908 (27.3%)), and atrial fibrillation (*n* = 146/908 (16.1%)). Therefore, none of the concomitant coronary artery disease cases required further intervention, and the Heart Team agreed upon conservative treatment. Ablation therapy was not performed in cases of concomitant atrial fibrillation in the studied cohort according to the inclusion criteria. Clinical baselines generally did not differ significantly between groups in the matched cohort (Table 2).

All baseline parameters were balanced between both groups. Therefore, the comparability of clinical baseline parameters between groups in the matched cohort was confirmed (Table 2).

### 3.3. Procedural Outcomes

Both techniques were safe without any major complications or even intraoperative deaths. The skin-to-skin time had a mean difference of 5 min and was significantly shorter in the sternotomy group (114.2 ± 28.7 min vs. 120 ± 61.7; *p* < 0.001). Procedures performed with the transaxillary approach were characterized by shorter x-clamp (41.0 ± 26.2 min vs. 50.5 ± 27.2 min; *p* < 0.001) and perfusion (63.0 ± 17.6 min; vs. 67.6 ± 14.3 min; *p* < 0.001) times. Figure 3 shows procedural times. However, x-clamp and perfusion-times did not differ significantly between groups within patients treated with sutured valves (Table 3).

Rapid deployment valves were used much more frequently in the transaxillary group (82.4% vs. 0.0%; *p* < 0.001). Moreover, the average labeled sizes of the implanted valves were larger in the transaxillary group (24.5 ± 2.8 mm vs. 23.4 ± 1.7 mm; *p* < 0.001). Mechanical substitutes were infrequently implanted in both groups.

The conversion rate to full sternotomy was 0.9% (*n* = 4) for the transaxillary group. The cannulation sites differed significantly between groups according to their inherent technical differences. Table 3 summarizes procedural data and details of the implanted prostheses.

### 3.4. Postoperative Outcomes

Given the low-risk status of the whole cohort, the postoperative course was mainly uneventful in both groups. Observed hospital mortality rates were 0.88% in the transaxillary and 1.54% in the sternotomy group, which was not significantly different (*p* = 0.5463). The observed mortality in the transaxillary group was lower than its EuroSCORE II predicted mortality (1.63 ± 1.1 %). However, the observed mortality in the sternotomy group almost matched its EuroSCORE II predicted mortality (1.49 ± 0.9 %).

Major postoperative morbidity due to stroke, transient ischemic attack (TIA), postoperative myocardial infarction, or renal failure needing temporary hemofiltration were generally infrequent, and their incidence did not differ between groups (Figure 4).

The main postoperative event was transient delirium development. This potential surrogate for deairing quality frequently occurred in the transaxillary (18.3%) and sternotomy (15.9%) groups and did not differ significantly between them (*p* = 0.855). However, significant intergroup differences were observed for the following variables.

In favor of the transaxillary approach, patients of the sternotomy group had significantly longer ventilation times (*p* < 0.001) and ICU stays (*p* < 0.001), translating into a significantly longer hospital stay (11.1 ± 6.5 days vs. 7.0 ± 5.1 days (*p* < 0.001)). Additionally, the average number of packed red blood cell (PRBC) transfusions was significantly lower in the transaxillary group (0.57 ± 1.6 vs. 0.82 ± 1.6; *p* = 0.040). In contrast, re-exploration for bleeding was more frequent in the transaxillary group (5.0% vs. 2.4%; *p* < 0.001), most likely for access-related reasons, with surgical bleedings more likely on the chest wall (*n* = 17/23 (7.4%)) than on the aortotomy or pulmonary vein (*n* = 3/23% (0.7%)). A further reason for re-thoracotomies in the transaxillary group was ostensible skin emphysema development (*n* = 3/23 (0.7%)). The incidence of access-related re-thoracotomies appeared to drop over time in the transaxillary group, potentially indicating an initial learning curve.

Wound healing abnormalities from exsudation to mediastinitis were comparable between groups, but their patterns differed significantly. In the sternotomy group, impaired wound healing was mostly due to sternal instability (*n* = 12/17 (2.6%)) and presternal dehiscences (*n* = 5/17 (1.1%)). Only one of 17 impaired wound healing cases in the sternotomy group could be treated in an ambulatory setting. The pattern of wound healing abnormalities in the transaxillary group was significantly different. Impaired wound healing in the transaxillary group comprised access (*n* = 7/18 (1.5%)) and cannulation (*n* = 11/18 (2.4%)) site problems. However, most cases (*n* = 14/18 (77.8%)) could be treated in an ambulatory setting. The overall rate of groin problems was 2.4% (*n* = 11/454). Table 4 summarizes the general postoperative outcomes comparing the clinical performance of both groups.

## 4. Discussion

An indisputable patient demand exists for minimally invasive methods causing less pain, quicker recovery, and improved cosmesis. However, how to meet this demand is still being debated. Transcatheter heart valve procedures increasingly challenge surgical techniques, and the role of minimally invasive techniques remains under-investigated.

While the pioneering Rao and Cosgrove teams attempted to develop minimally invasive surgical techniques in the 1990s, minimally invasive surgery remains underappreciated and has not advanced to the accepted standard of care [1,2,15,16]. For example, even >10 years after Rao (1993) and Cosgrove (1996) published their techniques, only a small proportion (4.7%) of all aortic valve procedures were not performed through full sternotomy in Germany in 2007 [15]. Moreover, even the innovative and bone-sparing RAT technique described by Joseph Lamelas was not a breakthrough success [8].

Only the emerging use of catheter-based procedures, introduced mainly after 2008, appeared to improve the situation [16]. At least, in Germany, the proportion of aortic valve surgeries not performed through sternotomy has risen to 36.8% in 2021 [16]. This development is the first important step to maintaining patient acceptance of surgical procedures.

Of course, transcatheter aortic valve procedures better meet the patient’s desire for the minimum invasiveness possible. However, future therapeutic decisions should predominantly rely on objective evidence than on subjective perception. Catheter-based procedures still have the unresolved shortcomings of paravalvular leakages, high pacemaker implantation rates, and undetermined implanted valve durability [17]. Therefore, transcatheter heart valve procedures remain restricted to inoperable, high-risk, or intermediate-risk patients after a thorough discussion in an interdisciplinary Heart Team [17]. In addition, attempts to extend catheter-based procedures to younger patients in the US were prevented by mounting evidence of increased reoperations after transcatheter heart valve procedures and an association with disproportionately high mortality [18,19,20,21,22].

Therefore, cardiac surgery will likely have a stake in the future of aortic valve therapies. Notably, surgical aortic valve replacement has an unarguable conceptual advantage: it is a genuine valve replacement, not a valve implantation that leaves the malfunctioning valve in place. However, patient acceptance and persuasion of the Heart Team discussion remain essential.

Consequently, the greatest possible adoption of minimally invasive techniques must be aimed for, but not at all costs. Patient safety is nonnegotiable, and all minimally invasive techniques must be as safe and effective as aortic valve surgery by full sternotomy.

The simplified “single incision—direct vision” transaxillary access proposed by our team comes without embellishment [10,11]. The technical description and first series presented a straightforward technique without an expensive or sophisticated setup [10,11]. This study sought to evaluate its clinical performance and efficacy compared to propensity-matched sternotomy patients.

Several studies have elaborated on MICS-AVR outcomes [23]. However, most included only a modest number of patients [23]. Compared to the absolute number of patients included in previous studies, the present series with 908 patients is one of the largest [23,24,25].

It appears evident that mortality rates are not increased in minimally invasive surgeries, indicating that the procedure is safe [24,26,27,28,29,30]. This study’s results are consistent with these findings. Remarkably, unlike in the sternotomy group, the observed mortality in the transaxillary group was lower than predicted by the EuroSCORE II. However, their better hospital survival was not statistically significant.

Unlike many studies describing longer x-clamp and perfusion times for minimally invasive surgeries, this study showed that both times were significantly shorter in the transaxillary group [23,31,32,33], likely due to the predominant use of rapid deployment valves in this series. These findings are supported by Borger et al., who found reduced x-clamp times when rapid deployment valves were used [34]. In this series, there was a tendency for sutureless valves for the following reasons. Since the use depended on the surgeon’s preference, possible reasons might be firstly, the high transverse aortotomy, which is easier to control through the minimally invasive access and secondly that larger distances from the aortic annulus to the chest wall could be accepted. Nevertheless, the transaxillary access can be used with every commercially available stented valve (Appendix A).

Though statistically significant, the mean procedure time differed by only 5 min between the transaxillary and sternotomy groups in this series. The observed mean skin-to-skin time of 120 min in the transaxillary group reflects this technique’s straightforward feasibility.

Major morbidities concerning neurologic events, myocardial infarction, and renal or respiratory failure did not differ significantly in this series. These observations are mainly consistent with several studies on MICS-AVR using different techniques [3,23,24,35]. The absence of an increase in the neurologic event rate in the transaxillary group must be emphasized since femoral cannulation was used in most cases. Additionally, the incidence of postoperative delirium did not differ significantly between the transaxillary and sternotomy groups. This finding is particularly interesting since we use the onset of delirium as a surrogate for deairing, which is traditionally an issue in MICS-AVR. In conclusion, MICS-AVR, and particularly transaxillary access, were not generally associated with increased major postoperative morbidity.

The main advantages of the transaxillary technique observed in this series were significantly shorter primary ventilation times and ICU stays. This quicker initial course, combined with a presumed faster mobilization due to the preserved and untouched osseous thorax, translated into a significant reduction in hospital stay by three days on average. These findings are inconsistently confirmed by the current literature, which mainly deals with hemisternotomies [23,31,32,33,36,37,38,39]. One possible explanation could be that transaxillary access does preserve the sternum and avoids any rib transection. Therefore, there are no limitations for postoperative mobilization, provided adequate pain management is given. Therefore, especially after hospital discharge, further apparent advantages of this bone and rib-preserving transaxillary technique are the absence of concerns about driving a car or lifting or supporting objects.

A minor drawback was the increased rate of re-exploration for bleeding. Despite some studies describing similar observations, most reported no difference in re-explorations or blood transfusions [23,40,41]. In this series, the causes for re-thoracotomies were access-related primarily, with small but persistent intercostal or muscle bleedings at the chest wall causing a hemothorax. These causes decreased over time, which could indicate an initial learning curve. Paradoxically, the higher rate of re-explorations was not associated with higher transfusion. In contrast, the transaxillary group had significantly fewer transfusions than the sternotomy group. One possible explanation for this finding is that bleedings in this series were generally minor. Nevertheless, as previously mentioned, the higher re-exploration rate did not translate into adverse clinical effects.

Another aspect is wound healing abnormalities. While the incidence of sternal complications did not differ significantly between the hemisternotomy and sternotomy approaches in the literature, an advantage exists favoring the RAT [23,25,31,38,42,43]. In this series, the overall incidence of wound abnormalities did not differ significantly between the transaxillary and sternotomy groups, but their patterns were completely different. Sternotomy procedures suffered from sternal instability and primarily severe wound complications requiring a hospital setup for adequate treatment. In contrast, most complications in the transaxillary group could be treated in an ambulatory setting. Most of the healing abnormalities were minor subcutaneous dehiscences at the cannulation site or, less frequently, at the surgical access site.

## 5. Conclusions

There exists no high-level evidence for the superiority of MICS-AVR compared to sternotomy. However, the question is whether superiority is really needed or is it sufficient that MICS-AVR can be performed in a comparable time frame and with a similar safety as a sternotomy. Transaxillary access for minimally invasive aortic valve surgery provides both: it can be performed as safely as aortic valve surgery by sternotomy with comparable procedural time. Additionally, no a priori contraindications exist after the learning curve, especially not by height and weight (Figure 2), except for severe lung adhesions.

This series also had obvious, but possibly less critical, advantages such as fewer postoperative transfusions, shorter ventilation times, and shorter ICU stays, translating into shorter hospital stays. General postoperative morbidity did not differ significantly. While the transaxillary group had better survival than the sternotomy group, the difference was not significant. The observed mortality rate in the transaxillary group was lower than predicted by EuroSCORE II.

For the patients themselves, the transaxillary approach offers some practical everyday advantages. The fact that the sternum and ribs remain untouched allows for a nearly normal physical exercise level in the early posthospital period. The most important benefit for patient acceptance is likely its superior cosmetic result, achievable with the same safety profile as a sternotomy.

## 6. Limitations

First, this series was a single-center retrospective study. Second, its lengthy observation period (over two decades) may bias its results due to inevitable changes in ICU management, patient blood management, operating surgeons’ experience, and other undocumented institutional changes over that time. Third, there was a significantly higher use of rapid deployment valves in the transaxillary group, which de facto represents a significant bias in the comparability of procedure times.

## Figures and Tables

**Figure 1 medicina-59-00160-f001:**
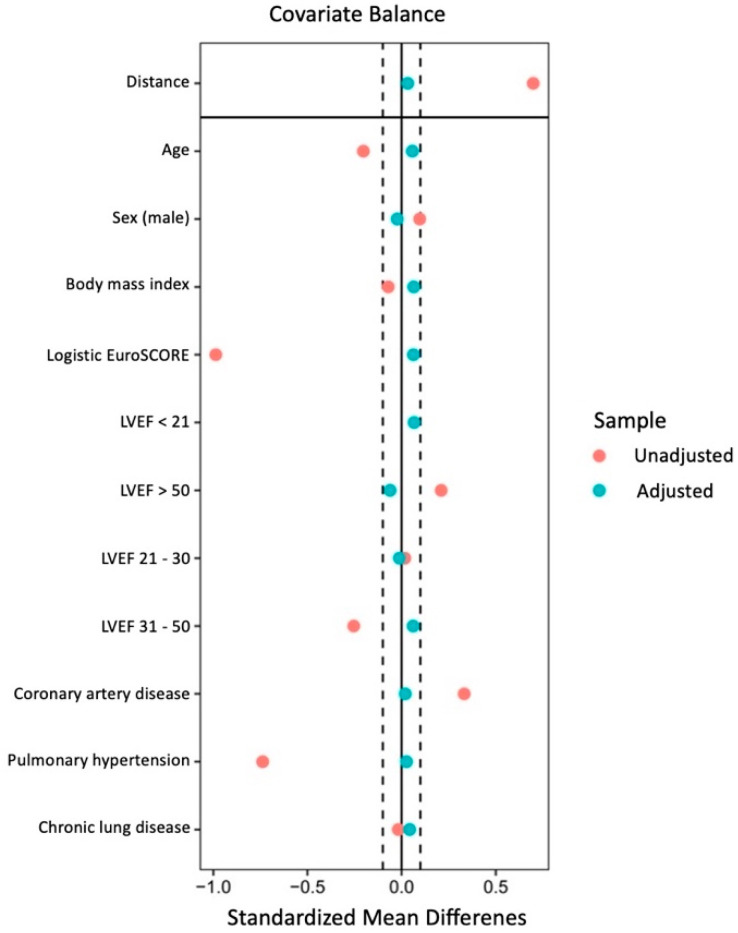
Covariate balance of the matching parameters before and after adjustment.

**Figure 2 medicina-59-00160-f002:**
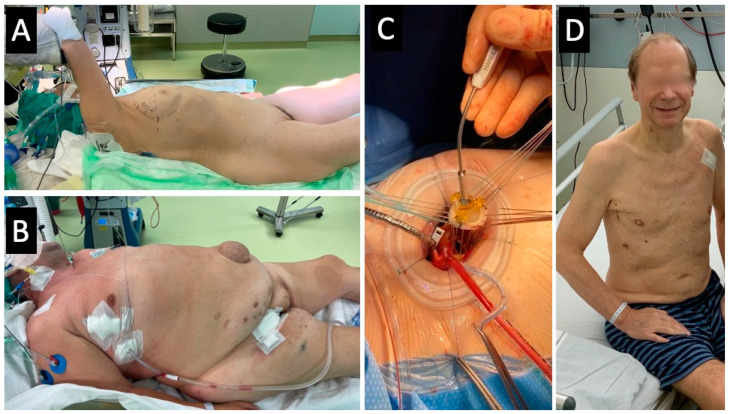
(**A**) Intraoperative setup in a slender female. (**B**) Immediate postprocedural aspect in a BMI 46 kg/m^2^ male. (**C**) Intraoperative aspect during implantation of a sutured bioprosthesis. (**D**) Postoperative cosmetic result in a male patient.

**Figure 3 medicina-59-00160-f003:**
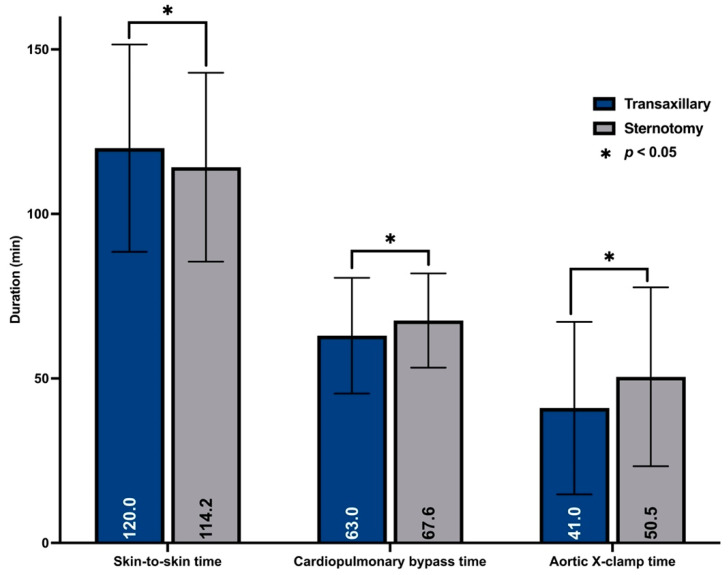
Procedural times.

**Figure 4 medicina-59-00160-f004:**
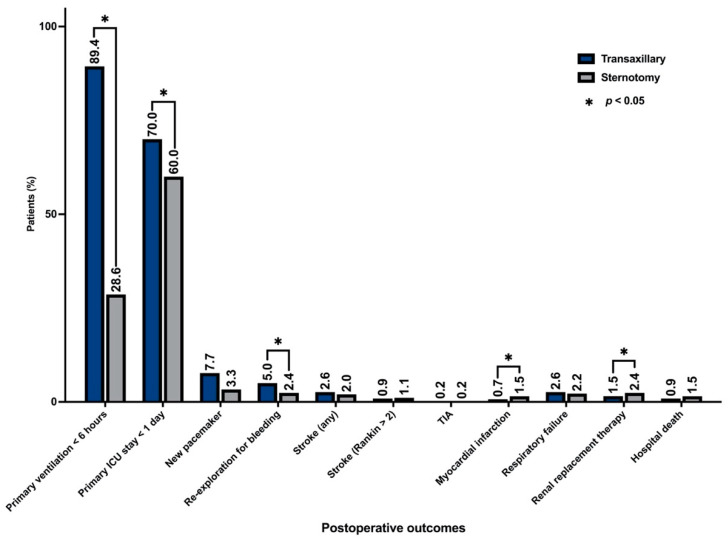
Main postoperative outcomes.

**Table 1 medicina-59-00160-t001:** Univariate analysis of balanced matching parameters after 1:1 propensity score matching.

	Propensity Matched Cohort (*n* = 908)		
	Transaxillary (*n* = 454)	Sternotomy (*n* = 454)	*p*
Age (years); mean ± SD	69.5 ± 18.3	69.2 ± 17.3	0.5293
BMI (kg/m^2^); mean ± SD	27.1 ± 8.3	27.4 ± 4.6	0.2994
LVEF by group: *n* (%)			
>50%	365 (80.4)	376 (82.8)	0.2904
31–50%	69 (15.2)	59 (13.0)	
21–30%	18 (4.0)	19 (4.2)	
<21%	2 (0.4)	0 (0.0)	
Pulmonary hypertension *, n (%)	13 (2.9)	11 (2.4)	0.837
Coronary artery disease, n (%)	126 (27.8)	122 (26.9)	0.8232
Chronic lung disease	32 (7.1)	27 (6.0)	0.5905
Logistic EuroSCORE (%); mean ± SD	4.1 ± 4.0	3.9 ± 3.8	0.3312

Abbreviations: BMI, body mass index; LVEF, left ventricular ejection fraction; * RVESP > 60 mmHg.

**Table 2 medicina-59-00160-t002:** Clinical baseline characteristics.

	Propensity Matched Cohort (*n* = 908)		
	Transaxillary (*n* = 454)	Sternotomy (*n* = 454)	*p*
Age (years); mean ± SD	69.5 ± 18.3	69.2 ± 17.3	0.5293
Sex (male): *n* (%)	281 (61.9)	286 (63.0)	0.7840
BMI (kg/m^2^); mean ± SD	27.1 ± 8.3	27.4 ± 4.6	0.2994
Diabetes mellitus: *n* (%)	125 (27.5)	132 (29.1)	0.9795
Coronary artery disease: *n* (%)	126 (27.8)	122 (26.9)	0.8232
LVEF (%): *n* (%)			
>50%	365 (80.4)	376 (82.8)	0.2904
31%–50%	69 (15.2)	59 (13.0)	
21%–30%	18 (4.0)	19 (4.2)	
<21%	2 (0.4)	0 (0.0)	
Pulmonary arterial hypertension: *n* (%)	13 (2.9)	11 (2.4)	0.837
Hemodialysis: *n* (%)	3 (0.7)	0 (0.0)	0.2492
Creatinine clearance (mL/min): mean ± SD	75.5 ± 28.0	75.9 ± 24.5	0.7792
Peripheral artery disease: *n* (%)	27 (6.0)	39 (8.5)	0.1000
Carotid artery stenosis >50%: *n* (%)	20 (4.4)	19 (4.2)	0.957
h/o TIA: *n* (%)	5 (1.1)	0 (0.0)	0.0618
h/o ischemic stroke: *n* (%)	24 (5.3)	21 (4.6)	0.7601
Atrial fibrillation: *n* (%)	88 (19.4)	58 (12.7)	0.0621
Smoker status: *n* (%)	52 (11.5)	49 (10.8)	0.795
Chronic lung disease	32 (7.1)	27 (6.0)	0.5905
NYHA class III or IV: *n* (%)	297 (65.4)	280 (61.7)	0.472
Logistic EuroSCORE (%); mean ± SD	4.1 ± 4.0	3.9 ± 3.8	0.3312
EuroSCORE II (%); mean ± SD	1.63 ± 1.1	1.49 ± 1.0	0.4953

Abbreviations: BMI, body mass index; h/o, history of; LVEF, left ventricular ejection fraction; TIA, transient ischemic attack.

**Table 3 medicina-59-00160-t003:** Procedural data.

	Propensity-Matched Cohort (*n* = 908)		
	Transaxillary (*n* = 454)	Sternotomy (*n* = 454)	*p*
Skin-to-skin time (min) mean ± SD	120.0 ± 31.5	114.2 ± 28.7	<0.001
Aortic x-clamp time (min) mean ± SD	41.0 ± 26.2	50.5 ± 27.2	<0.001
pts. with sutured prosthesis (min) mean ± SD	51.7 ± 14.4	50.5 ± 27.2	0.967
pts. with rapid deployment valve (min) mean ± SD	35.9 ± 25.9	n/a	n/a
CPB time (min): mean ± SD	63.0 ± 17.6	67.6 ± 14.3	<0.001
pts. with sutured prosthesis (min) mean ± SD	79.2 ± 20.9	67.6 ± 14.3	0.869
pts. with rapid deployment valve (min) mean ± SD	47.9 ± 14.8	n/a	n/a
Repeated x-clamping: *n* (%)	5 (1.1)	6 (1.3)	1.0000
Intraoperative death: *n* (%)	0 (0)	0 (0)	1.000
Conversion to sternotomy: *n* (%)	4 (0.9)	n/a	n/a
Major intraoperative complications ^†^: *n* (%)	0 (0)	0 (0)	1.000
Arterial cannulation: *n* (%)			<0.001
aorta	2 (0.4)	454 (100)	
femoral artery	450 (99.1)	0 (0)	
axillary artery	2 (0.4)	0 (0)	
Prosthesis type: *n* (%)			
biologic (sutured)	59 (13.0)	336 (74.0)	<0.001
biologic (rapid deployment)	374 (82.4)	0 (0.0)	
mechanic	21 (4.6)	118 (26.0)	
Labeled valve size (mm): mean ± SD	24.5 ± 2.8	23.4 ± 1.7	<0.001
Hemodynamic outcome (6th day TTE) *			
Pmax (mmHg); mean ± SD	24.1 ± 8.9	28.0 ± 6.7	<0.001
Pmean (mmHg); mean ± SD	12.9 ± 4.8	16.3 ± 5.3	<0.001
Paravalvular leakage (more than trace) **	3 (0.6)	2 (0.4)	1.000

^†^ Extracorporeal life support implantation was an unplanned procedure extension used in response to major intraoperative bleeding or other life-threatening intraprocedural complications; * Measured at the end of the procedure; ** paravalvular leak equal or > *I*° was not accepted by policy and revised. Abbreviations: CPB, cardiopulmonary bypass; TTE, transthoracic echocardiography.

**Table 4 medicina-59-00160-t004:** Postoperative outcomes.

	Propensity-Matched Cohort (*n* = 908)		
	Transaxillary (*n* = 454)	Sternotomy (*n* = 454)	*p*
Hospital death: n (%)	4 (0.88)	7 (1.54)	0.5463
Hospital stay (days); mean ± SD	7.0 ± 5.1	11.1 ± 6.5	<0.001
ICU stay (days): *n* (%)			
<1 day	318 (70.0)	272 (60.0)	<0.001
≤2 days	41 (9.0)	86 (18.9)	
≤3 days	47 (10.4)	45 (9.9)	
>3 days	48 (10.6)	51 (11.2)	
Ventilation time (hours): *n* (%)			
<6 h	406 (89.4)	130 (28.6)	<0.001
<12 h	35 (7.7)	302 (66.5)	
>12 h	13 (2.9)	22 (4.8)	
Respiratory failure ^†^: *n* (%)	12 (2.6)	10 (2.2)	0.8297
Postoperative transfusions:			
PRBC (pcs); mean ± SD	0.57 ± 1.6	0.82 ± 1.6	0.0401
Patients; n (%)	127 (27.9)	161 (35.5)	0.0423
Re-exploration for bleeding: *n* (%)	23 (5.0)	11 (2.4)	<0.001
Skin emphysema: *n* (%)	33 (7.3)	0 (0)	<0.001
Renal replacement therapy: *n* (%)	7 (1.54)	11 (2.4)	0.4762
Delirium: *n* (%)	83 (18.3)	72 (15.9)	0.8550
TIA: *n* (%)	1 (0.23)	1 (0.23)	1.000
Stroke: n (%)			
any	12 (2.6)	9 (2.0)	0.0891
Rankin >2	4 (0.9)	5 (1.1)	0.7910
Postoperative myocardial infarction: *n* (%)	3 (0.7)	7 (1.5)	0.0473
Impaired wound healing: *n* (%)	18 (3.9)	17 (3.7)	0.7341
surgical access site	7 (1.5)	17 (3.7)	0.0379
cannulation site (groin)	11 (2.4) *	0 (0)	<0.001
treatable in ambulatory care	14 (3.1)	1 (0.2)	<0.001
Mesenteric ischemia: *n* (%)	0 (0)	0 (0)	1.000
Permanent pacemaker implantation: *n* (%)	35 (7.7)	15 (3.3)	0.2203

^†^ Respiratory failure was defined as tracheotomy, reintubation, or primary ventilation for >24 h. Abbreviations: PRBC, packed red blood cells; TIA, transient ischemic attack; ICU, intensive care unit; * all lymphatic fistula/seroma.

## Data Availability

The data presented in this study are available on request from the corresponding author. The data are not publicly available due to ethical regulations.

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
