# Peer review of "Safety and Efficacy of the Transaxillary Access for Minimally Invasive Aortic Valve Surgery"

_medicina, 2023, doi:10.3390/medicina59010160_

Round 1

Reviewer 1 Report

Manuel Wilbring et al. performed a retrospective propensity matched study between patients who underwent transaxillary aortic valve replacement as treatment group vs sternotomy aortic valve replacement as control group. They conclude that the two procedure have similar results.

Nevertheless, I have some comments:

- The authors decided to include sternotomy patients from the pre-MICS- era to avoid bias. Even if it has a rationale, it has anyway bias regarding ICU length of stay, length of ventilation and overall length of stay, which were longer in the early 2000s than now, as of PRBC (acronym that has to be opened throughout the text) transfusion, since once transfusions were made with different criteria. Indeed, reexploration were more commonly performed in the Pre-MICS-era. Could the authors elaborate on this?

- The choice of sutureless prosthesis was made based on what criteria? Literature is debating over the actual durability of them. What do the authors think about it?

- In the discussion, the authors states that other studies are underpowered. Could they add in the methods part the description of the power of their sample?

- Is it possible to have some information regarding post operative results, such as presence of leaks? Do the authors have some follow up?

Some minor typos in the text (i.e. Line 175: NYHA III or for. I imagine the authors meant NYHA III or IV).

Author Response

Dear Reviewer,

thank you for your constructive comments. Please find below the point-to-point response. We hope that we adequately addressed your remarks and that the manuscript substantially has improved.

Sincerely

Manuel Wilbring on behalf of the authors

  1. The authors decided to include sternotomy patients from the pre-MICS- era to avoid bias. Even if it has a rationale, it has anyway bias regarding ICU length of stay, length of ventilation and overall length of stay, which were longer in the early 2000s than now, as of PRBC (acronym that has to be opened throughout the text) transfusion, since once transfusions were made with different criteria. Indeed, reexploration were more commonly performed in the Pre-MICS-era. Could the authors elaborate on this?

Response:That is completely right. In this point, we were between Scylla and Charibdis. Since we started with the MICS-program, we today reached a MICS-rate > 95%, which makes it impossible to find comparable patients. Especially in the transition period between 2015 and 2018, there likewise was a severe selection bias (only slim&beauty for MIC). For those reasons we added a limitations section elaborating on this point and we added the following paragraphs.

The transition period from 2015 to 2018 was excluded due to an obvious selection bias. During this period, the predominant access route for MICS-AVR was the RAT, with a series of 653 consecutive patients. This access was replaced by the transaxillary access beginning from 2019.

  1. d) Intensive Care Unit Setup

The intensive care unit (ICU) was a 24-bed, highly standardized cardiac surgery-only ICU. One anesthesiologist has continuously overseen its medical management since 1997, employed as a consultant at the Department of Cardiac Surgery. Therefore, all processes, especially ventilation, patient blood, and hemodynamic management, were characterized by high standardization and continuity with minimal variations over time.

  1. The choice of sutureless prosthesis was made based on what criteria? Literature is debating over the actual durability of them. What do the authors think about it?

Response:

We actually know about this issue. The perceval has for (from our point of view) unknown reasons the reputation of poor durability. But there exists some literature supporting excellent durability. This goes in line with our experience. Since 2014 we have been implanting >1.000 Perceval prostheses with a reoperation rate (for SVD) 0.3% over this 8-year period. We believe that in the initial phase before the change of the sizing chart a lot of valves were oversized, which lead to fluttering of some leaflets, lazy leaflets, valve thrombosis and high gradients. After change of the sizing chart, we (as a frequent user with >250 implants/year) do not see these problems anymore.

We can add this information if you want, but we more like the manuscript to focus on the surgical access (which is independently possible from the implanted valve). If we start the discussion about the perceval, we fear that intended focus on the access route will get into the background and the manuscript switches into a sutured/sutureless-discussion (which probably is endless). The attached supplemental video demonstrates that every stented valve is implantable through the transaxillary access.

 We took a corresponding paragraph in the discussion and in the methods section:

In this series, there was a tendency for sutureless valves for the following reasons. Since the use depended on the surgeon’s preference, possible reasons might be firstly, the high transverse aortotomy, which is easier to control through the minimally invasive access and secondly that larger distances from the aortic annulus to the chest wall could be accepted. Nevertheless, the transaxillary access can be used with every commercially available stented valve (Supplementary Video).

Prosthesis Choice: The prosthesis choice generally depended on the operating surgeon’s preference. Especially in the MICS era and in the MICS-AVR patients, detailed anatomical information for procedural planning was gathered preoperatively by a high-resolution full cardiac cycle computerized tomography scan of the heart and vessels (TAVI-CT protocol). This information helped determine the expected distances from the annular plane to the chest wall, aortic annulus size, and the expected implanted valve size. This information was used in the surgeon’s decision-making process. Since increasing evidence supports the durability of the mainly implanted rapid deployment valve, their use was unrestricted [12-14].

  1. In the discussion, the authors states that other studies are underpowered. Could they add in the methods part the description of the power of their sample? 

Response:

Since this is a retrospective study, an a priori power analyses makes little sense. The aim of an a priori power analysis is to guide sample size calculations on specific end points based on an assummed incidence of these endpoints. Since in a retrospective study the indicence of the endpoints is given (from the study population) and the number of patients and events is also given, there is little merit in statistical power calculations. In these cases the confidence intervals of the outcomes better describe the uncertainty in the estimates. To avoid leaving the reader with a false impression we have removed the following “underpowered” sentence from our revised manuscript.

  1. Is it possible to have some information regarding post operative results, such as presence of leaks? Do the authors have some follow up?

Response: We added the hemodynamic outcomes in Table 3. Since this a quite new technique, we do not yet have longer follow up data.

  1. Some minor typos in the text (i.e. Line 175: NYHA III or for. I imagine the authors meant NYHA III or IV).

Response:

These are corrected

Reviewer 2 Report

Minimally invasive cardiac surgery is a popular trend nowadays. We know, that we should use rather the term minimal access cardiac surgery, because the real invasion is sometimes larger. It is related to the longer cardiopulmonary bypass and cross-clamping times. On the other hand, avoiding sternotomy if it is possible allows quicker convalescence and good cosmetic effect. Facing the increasing number of transcatheter procedures it is the only way to compete for the patient.

The paper compares minimally invasive AVR using transaxillary access with conventional AVR through the median sternotomy. The authors of the manuscript invented the presented method and this paper is the continuation of their previous work. The indisputable advantage of this method is that it is a bone-sparing procedure.

The important limitation of the study results from the fact, that in one group 82% of patients received rapid deployment valves, while in the other group only 0,44%. Therefore comparing and matching the groups has little comparative significance. Especially not appropriate is to analyze the cross-clamp time or cardiopulmonary bypass time between both groups. Rapid deployment valves, which means biological prosthetic valves, makes difficult to compare with substantial number of mechanical valves. The control group is made of patients operated in pre-MICS era and I think it could be related with prolonged ventilation times as well.

Author Response

Dear Reviewer,

thank you for your constructive comments. Please find below the point-to-point response. We hope that we adequately addressed your remarks and that the manuscript substantially has improved.

Sincerely

Manuel Wilbring on behalf of the authors

Minimally invasive cardiac surgery is a popular trend nowadays. We know, that we should use rather the term minimal access cardiac surgery, because the real invasion is sometimes larger. It is related to the longer cardiopulmonary bypass and cross-clamping times. On the other hand, avoiding sternotomy if it is possible allows quicker convalescence and good cosmetic effect. Facing the increasing number of transcatheter procedures it is the only way to compete for the patient.

The paper compares minimally invasive AVR using transaxillary access with conventional AVR through the median sternotomy. The authors of the manuscript invented the presented method and this paper is the continuation of their previous work. The indisputable advantage of this method is that it is a bone-sparing procedure. 

The important limitation of the study results from the fact, that in one group 82% of patients received rapid deployment valves, while in the other group only 0,44%. Therefore comparing and matching the groups has little comparative significance. Especially not appropriate is to analyze the cross-clamp time or cardiopulmonary bypass time between both groups. Rapid deployment valves, which means biological prosthetic valves, makes difficult to compare with substantial number of mechanical valves. The control group is made of patients operated in pre-MICS era and I think it could be related with prolonged ventilation times as well.

Response:

We completely agree to this point. According Reviewer #1’s remark, we added corresponding paragraphs in the manuscript and a limitations section as well:

Methods:

The prosthesis choice generally depended on the operating surgeon’s preference. Especially in the MICS era and in the MICS-AVR patients, detailed anatomical information for procedural planning was gathered preoperatively by a high-resolution full cardiac cycle computerized tomography scan of the heart and vessels (TAVI-CT protocol). This information helped determine the expected distances from the annular plane to the chest wall, aortic annulus size, and the expected implanted valve size. This information was used in the surgeon’s decision-making process. Since increasing evidence supports the durability of the mainly implanted rapid deployment valve, their use was unrestricted [12-14].

Results:

However, x-clamp and perfusion-times did not differ significantly between groups with sutured valves (Table 3).

Discussion:

In this series, there was a tendency for sutureless valves for the following reasons. Since the use depended on the surgeon’s preference, possible reasons might be firstly, the high transverse aortotomy, which is easier to control through the minimally invasive access and secondly that larger distances from the aortic annulus to the chest wall could be accepted. Nevertheless, the transaxillary access can be used with every commercially available stented valve (Supplementary Video).

 Limitations:

First, this series was a single-center retrospective study. Second, its lengthy observation period (over two decades) may bias its results due to inevitable changes in ICU management, patient blood management, operating surgeons’ experience, and other undocumented institutional changes over that time. Third, there was a significantly higher use of rapid deployment valves in the transaxillary group, which de facto represents a significant bias in the comparability of procedure times.

Reviewer 3 Report

I reviewed the manuscript titled: “Safety and Efficacy of the Transaxillary Access for Minimally Invasive Aortic Valve Surgery” with great interest. Here are my comments:

1.       What does CS stand for ‘MICS_AVR’?

2.       Manuscript needs to be thoroughly assessed for grammatical errors. E.g. line 51: unarguably instead of unargued, proponent instead of protagonist in line 59 etc. There are many unconventional abbreviations used e.g. x-clamp and the discussion is plagued by grammatical errors and suboptimal scientific writing.

3.       I recommend presenting the continuous data in form of mean + Standard deviation instead of mean + 95% Confidence intervals

4.       Were there any absolute contraindications of transaxillary approach? Anatomy wise or for other reasons?

5.       I am assuming that the tests were two sided/tailed however authors should include it in methods under statistical analysis section

6.       In the propensity score matched sample, there is still statistically significant difference in incidence of coronary artery disease and chronic lung disease and peripheral arterial disease is almost statistically significant. Can the authors account for those factors and run another propensity match score analysis to make two groups more comparable?

7.       It seems like the lesser cross clamp and bypass time might be due to the rapid deployment valve use in transaxillary approach compared to regular valves used in sternotomy approach. Can the authors give mean CPB and cross clamp times for each type of valve separately in both approaches?

8.       Was aortic root enlargement procedure performed via transaxillary approach in any patients?

9.       Was there a change in ICU protocols in author’s institution about rapid extubation ad early ICU discharge after surgery? Could it have had an impact on short ICU and ventilator time in patients who underwent transaxillary approach?

10.   Seems like majority of sternotomy patients were extubated within 12 hrs. The 6-hour difference between two groups doesn’t seem to be a big practical difference.

11.   Another downside of the transaxillary approach is the wound and bleeding complications in groin cannulation site. Traditional sternotomy does not have this disadvantage. This should be mentioned in the manuscript discussion.

12.   Authors should report bleeding complication at groin cannulation site separately.

13. Authors should include the results of minimal invasive anterior thoracotomy and aortic valve replacement in discussion with appropriate citations. This approach is currently used in quite a few centers and is not abandoned as authors suggest in manuscript. 

Author Response

Dear Reviewer,

thank you for your constructive comments. Please find below the point-to-point response. We hope that we adequately addressed your remarks and that the manuscript substantially has improved.

Sincerely

Manuel Wilbring on behalf of the authors

  1. What does CS stand for ‘MICS_AVR’?

Response: This stands for “Minimally Invasive Cardiac Surgery – Aortic Valve Replacement” – we outlined this now better at the beginning of the text.

  1. Manuscript needs to be thoroughly assessed for grammatical errors. E.g. line 51: unarguably instead of unargued, proponent instead of protagonist in line 59 etc. There are many unconventional abbreviations used e.g. x-clamp and the discussion is plagued by grammatical errors and suboptimal scientific writing.

Response: The revised manuscript now is language checked by a native speaker. We hope that the readability has improved accordingly.

  1. I recommend presenting the continuous data in form of mean + Standard deviation instead of mean + 95% Confidence intervals

Response: We changed this into +/- Standard deviation

  1. Were there any absolute contraindications of transaxillary approach? Anatomy wise or for other reasons?

      Response: Actually there are no a-priori contraindications in a surgical AVR population. Severe pleural adhesions can make this access impossible (only one case out of this series with conversion to partial sternotomy). We added this information in the manuscript.

5. I am assuming that the tests were two sided/tailed however authors should include it in methods under statistical analysis section

Response: Yes, this is corrected. Two sided tests were used in the present work. This has been added to the manuscript in the statistical analysis section as follows. A two sided p < 0.05 was considered statistically significant

6. In the propensity score matched sample, there is still statistically significant difference in incidence of coronary artery disease and chronic lung disease and peripheral arterial disease is almost statistically significant. Can the authors account for those factors and run another propensity match score analysis to make two groups more comparable?

Response: The reviewer is correct that there were some variables that shown statistical significant differences between the control and treatment group, and the reviewer asks to take these differences into consideration in PSM model. However, we feel that this might add additional bias.

Aim of the propensity score matching is to correct for variables that may affect treatment allocation, in the present study conventional sternotomy or MICS AVR. However, treatment allocation in our study did now take place simultaneously. All patients before 2015 received conventional sternotomy and all patients after 2018 received MICS AVR. Since CAD and chronic lung disease did not affect treatment allocation we did not include these variables in the PSM matching model. Patients irrespective of CAD or chronic lung disease would have received the same treatment (according to the prespecified eras, MICSAVR after 2018 and conventional sternotomy before 2015).

 The in the PSM model we included clinical factors that we believe would have influenced the decision of treatment allocation like age, sex, BMI, logistic EuroSCORE, and stratified left ventricular ejection fraction (LVEF). We feel that if we include factors that did not influence treatment allocation only because these exhibit a statistical significant difference this might infuse additional bias in the results and we would like to avoid this. In addtion to this, these factors will have little influence on purely in hospital endpoints (as in the present study). Therefore we would like to avoid including in the PSM matching factors and variables that did not influence treatment allocation just because of between group differences which could be a chance finding.

7. It seems like the lesser cross clamp and bypass time might be due to the rapid deployment valve use in transaxillary approach compared to regular valves used in sternotomy approach. Can the authors give mean CPB and cross clamp times for each type of valve separately in both approaches?

      Response: We added this information and additionally added corresponding paragraphs in the discussion / limitations section.

  1. Was aortic root enlargement procedure performed via transaxillary approach in any patients?

      Response: From the technical aspect, a patch enlargement of the root is possible - more we did not try. In our series, each patient had a CT scan prior the procedure for measurement distances and the annular size. Concerning the annular diameter, the implantable valve and expected hemodynamic outcome were predicted. During the enrollment time we actually did not have any unplanned Manougian or Y-plasty.

  1. Was there a change in ICU protocols in author’s institution about rapid extubation ad early ICU discharge after surgery? Could it have had an impact on short ICU and ventilator time in patients who underwent transaxillary approach?

      Response: Our ICU is lead by one person since 1997 and everything is highly standardized. All protocols are very constant over time. Anyhow, a change of habits over time is not unrealistic. We added a corresponding part in the Limitation section. Additionally, the ICU-setup is explained in the Methods section.

  1. Seems like majority of sternotomy patients were extubated within 12 hrs. The 6-hour difference between two groups doesn’t seem to be a big practical difference.

      Response: You are true. This is not really a big deal. The key message of our manuscript should be that MICS-AVR is at least as safe as sternotomy. We think non-inferior is sufficient in this case and discussed it that way. Actually - and even in the whole literature – there exists no real “hard” reason for MICS. To be honest, for us it is a tool for patient satisfaction, for Heart Team discussion as well as promotion. The most important thing to us is that MICS-AVR must be as safe and in the same time frame.

  1. Another downside of the transaxillary approach is the wound and bleeding complications in groin cannulation site. Traditional sternotomy does not have this disadvantage. This should be mentioned in the manuscript discussion.

      Response: To this point of time we fortunately experienced no bleeding complications in the groin since we performed a cut down in every patient. Percutaneous techniques are possible, but we actually do not perform them in the MICS-AVR. The reason for not doing percutaneous are for us, the lacking possibility for intraoperative proof of successful closure (we know from TAVR there is a rate of vessel occlusions) and the mentioned bleeding complications. All experienced – and mentioned – complications in the groin were wound healing problems (mostly due to lymphatic fistulas; overall rate of groin problems were 2.4%).

  1. Authors should report bleeding complication at groin cannulation site separately.

      Response: To this point of time we fortunately experienced no bleeding complications in the groin.

  1. Authors should include the results of minimal invasive anterior thoracotomy and aortic valve replacement in discussion with appropriate citations. This approach is currently used in quite a few centers and is not abandoned as authors suggest in manuscript. 

      Response: We would like to focus our manuscript more on the feasibility and safety of the transaxillary access compared to the standard sternotomy approach (which is used in more than 60% of all AVR’s in Germany in 2021). But on the side: we are preparing a manuscript comparing the RAT performed at our institution (n=653) with the transaxillary cases (in advance: no real differences). We added the following part in our methods section: The transition period from 2015 to 2018 was blinded due to an obvious selection bias. During this period, the predominant access route for MICS-AVR was the right anterior minithoracotomy (RAT) with a series of 653 consecutive patients. This access was completely left in 2019 in favor of the transaxillary access.

Round 2

Reviewer 1 Report

Authors have answered to all my comments. Thank you

Author Response

Dear Reviewer,

thank you very much for your help to improve the manuscript.

Best

Manuel Wilbring

Reviewer 3 Report

I thank the authors for the changes they have made according to my suggestions.

1. I would like to point out that although the comorbidities did not affect patient selection in the study, it may still have impact on post-operative outcomes like length of stay, mortality etc. I do not believe that including these variables will add bias. It will only refine the results. The way the study is created, it seems that the authors moved from the sternotomy to the axillary access regardless of any factors present. With that and the logic presented in the authors response, you may not have to account for any factors which clearly is an erroneous method to do the analysis. I still recommend adding the pertinent comorbidities to the analysis just to get the results between two comparable groups. for example, must by having increased pulmonary hypertension patient may need additional ICU treatments and vetilator and that may require longer stay in ICU. The incidence of pulmonary hypertension is higher in sternotomy group and therefore they may have higher length of stay associated with the comorbidities and not necessarily the approach.

2. Reoperation due to bleeding is higher in the  transaxillary approach. Having a higher incidence of complication, is this approach really equally safe compared to sternotomy?

3. The groin seroma/lymphatic complications should be added in the complications/outcomes table.

4. There appears to be 0.23 unit of transfusion benefit in transaxillary group. I would argue that it probably equals to less than 100 cc of blood transfusion. Alog with this metric, it would be beneficial to know what the incidence of blood transfusion was in each group. With more bleeding and takebacks in transaxillary group, was the incidence of blood transfusion higher in this group? if so, does it truly mean the transaxillary approach had less transfusion requirement?

Author Response

Dear Reviewer,

thank you for your comments. Please find below the point-to-point response. In the manuscript our changes made are highlighted red.

We hope that we adequately addressed your remarks.

Sincerely

Manuel Wilbring on behalf of the authors

  1. I would like to point out that although the comorbidities did not affect patient selection in the study, it may still have impact on post-operative outcomes like length of stay, mortality etc. I do not believe that including these variables will add bias. It will only refine the results. The way the study is created, it seems that the authors moved from the sternotomy to the axillary access regardless of any factors present. With that and the logic presented in the authors response, you may not have to account for any factors which clearly is an erroneous method to do the analysis. I still recommend adding the pertinent comorbidities to the analysis just to get the results between two comparable groups. for example, must by having increased pulmonary hypertension patient may need additional ICU treatments and vetilator and that may require longer stay in ICU. The incidence of pulmonary hypertension is higher in sternotomy group and therefore they may have higher length of stay associated with the comorbidities and not necessarily the approach.

Response: We adapted the matching criteria according your suggestions.

  1. Reoperation due to bleeding is higher in the  transaxillary approach. Having a higher incidence of complication, is this approach really equally safe compared to sternotomy?

Response: In our point of view definitely yes. Bleedings mostly were minor bleedings. We pursued a very aggressive revision policy. According to the minor bleedings, the second round in the OR did not translate into adverse clinical outcome. On the contrary, MICS had lower mortality, less transfusions, shorter ventilation, shorter ICU-stay and shorter hospital stay. Therefore, we believe it is more safe.

  1. The groin seroma/lymphatic complications should be added in the complications/outcomes table.

Response: We added a corresponding explanation in the table legend.

  1. There appears to be 0.23 unit of transfusion benefit in transaxillary group. I would argue that it probably equals to less than 100 cc of blood transfusion. Alog with this metric, it would be beneficial to know what the incidence of blood transfusion was in each group. With more bleeding and takebacks in transaxillary group, was the incidence of blood transfusion higher in this group? if so, does it truly mean the transaxillary approach had less transfusion requirement?

Response: We added this information in the legend. The transaxillary patients had less risk for transfusion.